# Improved prediction of smoking status via isoform-aware RNA-seq deep learning models

**Zifeng Wang**[1], **Aria Masoomi**[1], **Zhonghui Xu**[2], **Adel Boueiz**[2,3], **Sool Lee**[2], **Tingting Zhao**[1], **Russell Bowler**[4], **Michael Cho**[2,3], **Edwin K. Silverman**[2,3], **Craig Hersh**[2,3], **Jennifer Dy**[1], **Peter J. Castaldi**[2,5]*

**1** Department of ECE, Northeastern University, Boston, Massachusetts, United States, **2** Channing Division of Network Medicine, Brigham and Women's Hospital, Boston, Massachusetts, United States, **3** Division of Pulmonary and Critical Care Medicine, Brigham and Women's Hospital, Boston, Massachusetts, United States, **4** Division of Pulmonary and Critical Care Medicine, National Jewish Health, Denver, Colorado, United States, **5** Division of General Internal Medicine and Primary Care, Brigham and Women's Hospital, Boston, Massachusetts, United States

* repjc@channing.harvard.edu

**Data Availability Statement:** The data underlying the results presented in the study have been deposited in GEO (GSE158699).

## Abstract

Most predictive models based on gene expression data do not leverage information related to gene splicing, despite the fact that splicing is a fundamental feature of eukaryotic gene expression. Cigarette smoking is an important environmental risk factor for many diseases, and it has profound effects on gene expression. Using smoking status as a prediction target, we developed deep neural network predictive models using gene, exon, and isoform level quantifications from RNA sequencing data in 2,557 subjects in the COPDGene Study. We observed that models using exon and isoform quantifications clearly outperformed gene-level models when using data from 5 genes from a previously published prediction model. Whereas the test set performance of the previously published model was 0.82 in the original publication, our exon-based models including an exon-to-isoform mapping layer achieved a test set AUC (area under the receiver operating characteristic) of 0.88, which improved to an AUC of 0.94 using exon quantifications from a larger set of genes. Isoform variability is an important source of latent information in RNA-seq data that can be used to improve clinical prediction models.

## Author summary

Predictive models based on gene expression are already a part of medical decision making for selected situations such as early breast cancer treatment. Most of these models are based on measures that do not capture critical aspects of gene splicing, but with RNA sequencing it is possible to capture some of these aspects of alternative splicing and use them to improve clinical predictions. Building on previous models to predict cigarette smoking status, we show that measures of alternative splicing significantly improve the accuracy of these predictive models.

**Funding:** The project described was supported by Award Numbers U01 HL089897, U01 HL089856, R01 HL124233, and R01 HL147326 from the National Heart, Lung, and Blood Institute and the FDA Center for Tobacco Products (CTP). The content is solely the responsibility of the authors and does not necessarily represent the official views of the NIH or the Food and Drug Administration. COPDGene is also supported by the COPD Foundation through contributions made to an Industry Advisory Board comprised of AstraZeneca, Boehringer-Ingelheim, Genentech, GlaxoSmithKline, Novartis, Pfizer, and Sunovion. The funders had no role in study design, data collection and analysis, decision to publish, or preparation of the manuscript.

**Competing interests:** I have read the journal's policy and the authors of this manuscript have the following competing interests: Michael Cho has received grant support from GlaxoSmithKline and Bayer, consulting fees from Genentech and AstraZeneca, and speaking fees from Illumina. Edwin Silverman received grant support from GlaxoSmithKline and Bayer. Peter Castaldi has received grant support from GlaxoSmithKline and personal fees from GlaxoSmithKline and Novartis. Craig Hersh reports grant support from Boehringer-Ingelheim, Novartis, Bayer and Vertex.

## Introduction

Smoking is the most important environmental risk factor for a wide range of diseases including cardiovascular disease, lung cancer, and chronic obstructive pulmonary disease (COPD). Smoking increases the risk for these diseases through a variety of mechanisms including selective activation and repression of distinct aspects of the inflammatory response [1].

A meta-analysis of blood gene expression arrays from >5,000 current and never smokers identified 1,270 smoking-associated differentially expressed genes that were significantly enriched in immune-related processes including T-cell activation [2]. However, it is difficult to characterize the effects of cigarette smoking on splicing and differential isoform usage due to technical challenges in measuring isoform expression levels. Using RNA-seq combined with novel isoform reconstruction algorithms, we have shown that smoking causes widespread differential isoform and exon usage in addition to overall gene-level expression changes [3].

A five gene expression-based predictive model for smoking was previously proposed by Beineke et. al [4] with an AUC of 0.82, indicating that there is still room for improvement in predictive performance for expression-based prediction tools for current smoking status. High throughput measurements of gene expression in biological samples have been shown to capture information relevant to complex biological processes such as cell cycle [5], stress response [6], and medical disease states [7]. Gene expression-based multigene predictive models have achieved a level of performance that has resulted in their regular use in medical decision making, most notably in early stage breast cancer [8], but this level of precision has not yet been attained in many other areas of clinical practice. More recent research has applied neural networks to gene expression data in biomedical domains [9], achieving superior performance relative to other machine learning methods in some cases [10], though this is not a universal finding [11]. While gene expression microarrays were first used for genomewide transcriptomics profiling, massively parallel high-throughput RNA sequencing (RNA-seq) is now the standard, and one of the benefits of RNA-seq is that it can directly measure exon expression and detect junctional reads (i.e. RNA-seq reads spanning exons) which allows for estimation of transcript isoforms. It has been shown that the additional information that RNA-seq provides on alternative splicing allows for more sensitive detection of transcriptomic differences between cancer subtypes, but this information did not necessarily lead to improved prediction of clinical outcomes [12], suggesting that there may be latent information in RNA-seq data related to splicing that may require novel modeling approaches to better utilize this information.

Using blood RNA-seq data from 2,557 subjects in the COPDGene Study, we explored the relative utility of expression measures at the gene, exon, and isoform level using deep learning models [13] tailored specifically to account for patterns of alternative splicing induced by smoking. We hypothesized that since smoking alters patterns of exon and isoform usage, greater predictive accuracy could be obtained by using exon and isoform-level quantifications to predict smoking status.

## Materials and methods

### Ethics statement

Informed consent was obtained for all study subjects. The COPDGene Study received institutional approval from the Institutional Review Boards of all participating institutions as included in S2 Text.

## Subject enrollment and data collection

This study includes 2,557 subjects from the COPDGene Study. COPDGene has been previously described [14]. Self-identified non-Hispanic whites (NHW) and African Americans (AA) between the ages of 45 and 80 years with a minimum of 10 pack-years lifetime smoking history were enrolled at 21 centers across the United States. COPDGene conducted two study visits approximately five years apart, and the ten-year visits are being completed. Starting at the second study visit, complete blood count (CBC) data and PAXgene RNA tubes were collected. Smoking history was ascertained by self-report. Participants defined as current smokers answered yes to the question "Do you smoke cigarettes now (as of one month ago?)". Institutional review board approval and written informed consent were obtained.

## Total RNA extraction

Total RNA was extracted from PAXgene Blood RNA tubes using the Qiagen PreAnalytiX PAXgene Blood miRNA Kit (Qiagen, Valencia, CA). The extraction protocol was performed either manually or with the Qiagen QIAcube extraction robot according to the company's standard operating procedure.

## cDNA library construction and sequencing

Globin reduction and cDNA library preparation for total RNA was performed with the Illumina TruSeq Stranded Total RNA with Ribo-Zero Globin kit (Illumina, Inc., San Diego, CA). Library quality control included quantification with picogreen, size analysis on an Agilent Bioanalyzer or Tapestation 2200 (Agilent, Santa Clara, CA), and qPCR quantitation against a standard curve. 75 bp paired end reads were generated on Illumina sequencers. Samples were sequenced to an average depth of 20 million reads. All sequenced samples had RNA integrity number (RIN) > 6.

## Sequencing read alignment, quality control and expression quantification

Reads were trimmed of TruSeq adapters using Skewer with default parameters [15]. Trimmed reads were aligned to the GRCh38 genome using the STAR aligner in 2-pass mode [16]. We used these arguments for the first pass: STAR –runThreadN 8 –outSAMunmapped Within –outSAMstrandField intronMotif –outSJfilterReads Unique –outSJfilterCountUniqueMin 100 1 1 1. For the second pass, we provided splice junctions from the first pass with these additional arguments: –outSAMtype BAM SortedByCoordinate –limitSjdbInsertNsj 10000000 –chimSegmentMin 10 –sjdbFileChrStartEnd SJ.out.tab. Quality control was performed using the FastQC and RNA-SeQC programs [17]. Samples were included for subsequent analysis if they had >10 million total reads, >80% of reads mapped to the reference genome, XIST and Y chromosome expression was consistent with reported gender, <10% of R1 reads in the sense orientation, Pearson correlation ≥ 0.9 with samples in the same library construction batch, and concordant genotype calls between variants called from RNA sequencing reads and DNA genotyping.

Gene and transcript gene transfer file (GTF) annotation was downloaded from Biomart Ensembl database (Ensembl Genes release 94, GRCh38.p12 assembly) on October 21, 2018. We further derived exonic parts GTF annotation by breaking exons into disjoint parts sharing a common set of transcripts within a single gene. Sequencing read counts on gene and exonic part level genomic features were obtained from featureCounts function in Rsubread package (v1.32.2). Isoform level expression quantifications were derived using the Salmon program

(v0.12.0) and the tximport package (v1.10.0). The gene, isoform, and exon count data used for this analysis are available in GEO [26, 27] (GSE 158699).

## Filtering, normalization and covariates adjustment

Genomic features (genes, isoforms or exonic parts) were filtered for both features that had very low and very high expression. The filter used to remove low expressed features was to remove features where the average counts per million (CPM) was < 0.2 or the feature was not expressed at a CPM > 0.5 in at least 50 subjects. Extreme highly expressed features were defined as features attaining a CPM > 50,000 in at least one but fewer than 50 subjects. Differences in sequencing depth and RNA library composition between subjects were normalized using the TMM procedure from edgeR package (v3.24.3). Counts were transformed to log2 CPM values and quantile-normalized to further remove systematic noise. To avoid overfitting, we limited our set of genes to those contained within a set of 1,270 smoking-associated differentially expressed genes that had been identified in a previous study using samples that did not overlap with this study [2].

To assess the impact of demographic covariates of age, sex and body mass index, we generated covariates-adjusted expression data for reevaluating our model performance. Specifically, we fitted a linear model of the expression data using the demographic covariates and smoking status as explanatory variables, and removed the covariates effect while retaining the main effect of smoking. The adjustment procedure was performed using removeBatchEffect function from the limma package (v3.38.3).

## Data usage and model validation

We analyzed blood RNA-seq data from 2,557 subjects in the COPDGene Study. We randomly split the data into training, validation, and testing sets containing 1637, 407, 513 subjects respectively. Model optimization and hyperparameter tuning was performed in the training data using 5-fold cross-validation. A small set of high-performing models were further evaluated in the validation dataset, and the testing data were used only for evaluation of the final set of models after all parameters and hyperparameters were fixed. The testing data was held by a separate analysis group using a different computer system to avoid any possibility of inadvertent use of test data in the model building process.

## Model training

For all experiments, we train each model for 40 epochs with batch size 256 using Adam optimizer [18] with learning rate 0.0003, $\beta_1 = 0.9$, $\beta_2 = 0.999$ and a dropout [19] rate of 0.2. We employ the early stopping strategy during training. Specifically, we stop training and get the best performing model on the validation set so far when there is no increase in validation accuracy for 10 epochs. We set the weight of the L1 constraint used in the Feature Selection layer to be 0.0005. Unless otherwise specified, model layers were fully connected and ReLU nonlinear activation function were used. All the deep learning models were implemented in TensorFlow (v12.0.0) and Keras (v2.2.4.). A visualization of all network architectures can be found in S4 Fig. All network definitions, network weights and code, as well as additional files required for reproducing our experiment results are available at https://doi.org/10.5281/zenodo.5136729.

All experiments are conducted on a single NVIDIA GTX 1080Ti GPU. The training process related to Beineke models ranged from 13 seconds to 17 seconds for 40 epochs. The training process related to a larger feature set ranged from 58 seconds (the base model) to 252 seconds (model with IML and FSL) for 40 epochs.

To identify high-performing model architectures, we adopted a layer-by-layer incremental search strategy. We first explored the optimal number of nodes for a single layer network by performing grid search. We evaluated from 2 to 512 nodes in the first layer, increasing by a factor of 2 at each step, i.e., 2, 4, 8, . . ., 512. The number of nodes at each layer was selected based on cross-validation performance, and then an additional layer was added using the same grid search strategy for node number with the constraint that each subsequent layer would have fewer nodes than the preceding layer. Note that we reinitialize the weights of the previous layers when searching for the current layer. This process was repeated until no further gain in performance was achieved.

### Implementation of isoform map and feature selection layers

To incorporate prior knowledge regarding the relationship of exons to transcript isoforms, we implemented an Isoform Map Layer (IML) which takes exon features as input and outputs estimated isoform features. This specially-designed layer is based on a standard fully-connected layer with weight $\mathbf{W}$. This layer encodes known exon to isoform relationships in a binary relationship matrix $\mathbf{R}$ such that if exon $i$ is contained within isoform $j$, we set $\mathbf{R}_{ij} = 1$, otherwise $\mathbf{R}_{ij} = 0$. This layer takes the relationship matrix to perform element-wise multiplication with the learnable weight matrix $\mathbf{W}$. Thus, only canonical exon to isoform relationships can contribute to the final model. Exon to isoform relationships were obtained from the Ensemble v94 GTF file. Analogously, we can devise an exon-to-gene mapping layer, where exons are connected to the genes they are associated with, in the same way. We refer to this as the Gene Map Layer (GML).

To enhance interpretability, we included in some models a Feature Selection Layer (FSL) that associates every input feature with a non-negative learnable weight using an L1 constraint and outputs a reweighted feature vector of the same size as the input feature vector. Since the weights are non-negative, they can be considered to represent each feature's importance with respect to smoking status prediction, and the L1 constraint is meant to improve the generalizability of the model.

### Baseline models, model comparisons and interpretation

To assess the effectiveness of our method, we compared our method against the current method proposed by Beineke et al., which is a logistic regression model using the following five genes: *CLDND1*, *LRRN3*, *GOPC*, *LEF1*, *MUC1*. We apply the Beineke model on our data by exploring logistic regression with exon, isoform and gene inputs considering only these five genes. For models evaluating the full set of genes, we trained elastic net models as a baseline for comparison with the weights for the L1 and L2 norms set at 0.0005. We obtain the optimal set of parameters of elastic net by conducting grid search and find out the best performance on the validation set. All statistical tests of model performance were analyzed using data from the test set and performed using R version 3.6. Direct comparisons between models were performed using the deLong test implemented in the pROC package. Gene set functional enrichment analysis was performed for genes ranked in the top 20% of feature importance scores generated by the DeepExplain [20] framework with saliency maps [21]. We used the TopGO package [22] to compute the gene set enrichment p-values for Gene Ontology pathways using the 'weight01' algorithm with Fisher's exact test statistic.

### Serum cotinine measurement

Cotinine measurements were obtained from plasma through metabolomic profiling using the Metabolon Global Metabolomics Platform (Durham, NC, USA). The data were further

normalized to remove batch effects. Samples with undetectable cotinine levels were assigned a value of zero. COPDGene metabolomic data is available at the NIH Common Fund's National Metabolomics Data Repository website, the Metabolomics Workbench, https://www.metabolomicsworkbench.org (Study ID ST001443). Predictive accuracy of serum cotinine for smoking status was assessed with ROC curves applied to measured cotinine values.

## Results

RNA-seq data from 2,557 current and former smokers in the COPDGene Study were used to develop and test the predictive models. Data were randomly split into training, validation, and testing data. The use of data for model training, selection, and testing are described in Fig 1. The characteristics of the subjects in these datasets are shown in Table 1.

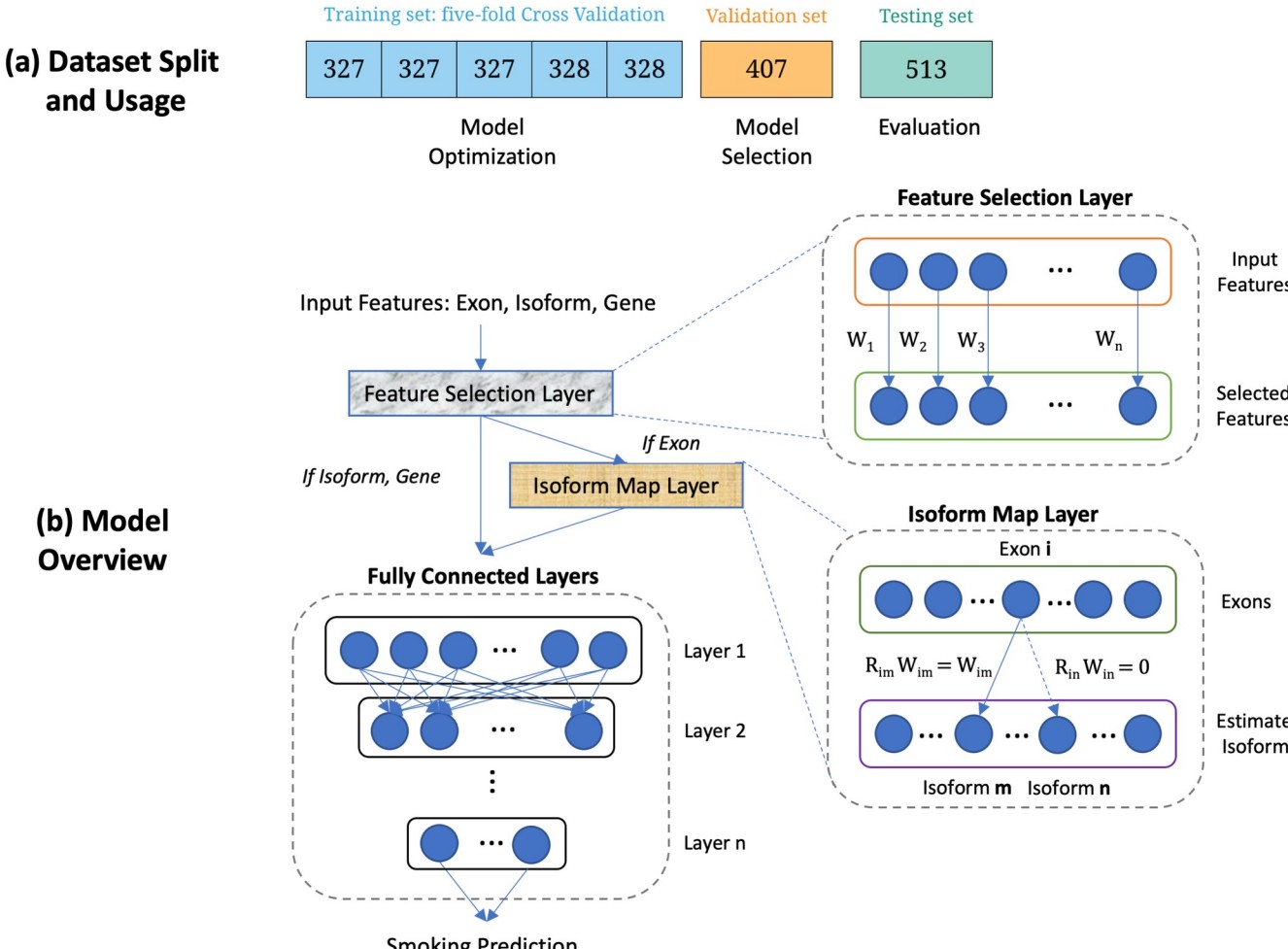

**Fig 1. Visual abstract.** (a) Dataset split and usage. The number in each cell represents the number of subjects. The training set is equally split into 5 folds for deep learning model optimization (cross-validation for tuning the hyperparameters and architecture search in a deep learning model). The validation set is used to select the optimal model and the testing set is held out for performance evaluation. (b) Model overview. Our model consists of a Feature Selection Layer (FSL), an Isoform Map Layer (IML) (if the input feature is exon) and standard fully connected layers. FSL associates each input feature with a non-negative learnable weight, which represents the importance of features with respect to smoking status. IML encodes exon to isoform relationships via a binary matrix $R$, such that if exon $i$ is contained within isoform $j$, we set $R_{ij} = 1$, otherwise $R_{ij} = 0$. By (element-wise) multiplying $R_{ij}$ with corresponding learnable weights $W$, we only consider canonical exon to isoform relationships.

**Table 1. Characteristics of subjects.**

|  | Training | Validation | Testing | P-value |
|---|---|---|---|---|
| Number of subjects | 1637 | 407 | 513 |  |
| Age, years | 65.4 (58.6, 71.9) | 65.6 (58.4, 71.3) | 65.4 (58.6, 71.7) | 0.2 |
| Sex, %males | 51.1% | 55.8% | 49.9% | 0.2 |
| Race, %non-Hispanic whites | 74.3% | 74.9% | 77.8% | 0.3 |
| BMI | 28.1 (24.5, 32.3) | 28.1 (25.0, 32.1) | 27.9 (25.1, 32.2) | 0.4 |
| Smoking pack-years | 40.0 (28.0, 54.8) | 40.0 (26.9, 52.7) | 40.0 (28.0, 57.9) | 0.8 |
| Current smokers, % | 35.4% | 35.4% | 35.5% | 0.9 |
| $FEV_1$, %predicted | 81.5 (62.7, 95.2) | 84.1 (66.8, 97.3) | 82.2 (63.7, 96.2) | 0.08 |
| $FEV_1$/FVC | 0.71 (0.61, 0.78) | 0.72 (0.62, 0.78) | 0.72 (0.62, 0.79) | 0.7 |
| COPD case status, % | 31.7% | 28.6% | 29.3% | 0.3 |

All values are from the COPDGene visit 2. BMI: Body mass index; FEV1: Forced expiratory volume in 1 second; FVC: Forced vital capacity; GOLD: Global Initiative for Chronic Obstructive Lung Disease; COPD case status defined as subjects with GOLD spirometric grade $\geq$ 2. Variables are expressed as medians and interquartile ranges (25th to 75th percentiles) for continuous variables, and percentages for categorical variables. P-values are obtained using the Kruskal-Wallis test for the continuous variables and chi-square test for the proportions.

## Validation and further development of the Beineke model using exon and isoform level data

A microarray and RT-PCR based five gene expression model for smoking has been previously developed and shown to have a test set AUC of 0.82 [4]. To externally validate this model and establish a performance benchmark in our dataset, we constructed an initial set of models using gene, exon, and isoform expression from this set of genes. One of the genes, *MUC1* was expressed in our data at levels below our filtering threshold. We confirmed that this gene is also expressed in very low levels in whole blood RNA samples from the Genotype Tissue Expression Project, and subsequently based our models on the other four gene expression values. A logistic regression model using these four genes had an AUC of 0.76 and 0.78 in our validation and testing data (Table 2). We then trained two additional logistic regression models using exon counts and Salmon estimated isoform quantifications from these genes. As shown in Fig 2, the prediction performance in both validation and testing datasets was improved using both isoform (p = 0.002) and exon level (p<0.001) quantifications, and exon data outperformed Salmon estimated isoform data (p = 0.002). Notably, the best performing models

**Table 2. Predictive performance of modified Beineke models using gene, isoform and exon-level expression data.**

|  | Val—Accuracy | Val—AUC | Test—Accuracy | Test—AUC |
|---|---|---|---|---|
| Gene | 0.698 | 0.758 | 0.743 | 0.780 |
| Isoform | 0.757 | 0.827 | 0.774 | 0.828 |
| Exon | 0.801 | 0.859 | 0.808 | 0.869 |
| Exon, GML-GTF | 0.771 | 0.807 | 0.789 | 0.811 |
| Exon, GML-GTF, FSL | 0.776 | 0.805 | 0.741 | 0.796 |
| Exon, IML-GTF | **0.828** | 0.876 | 0.825 | 0.870 |
| Exon, IML-GTF, FSL | **0.828** | **0.889** | **0.838** | **0.875** |

Val: validation data. AUC: area under the receiver operating characteristic. IML-GTF: Isoform Map Layer containing information from Ensembl GTF file. GML-GTF: Gene Map Layer containing information from Ensembl GTF file. FSL: Feature Selection Layer. Best results are shown in bold.

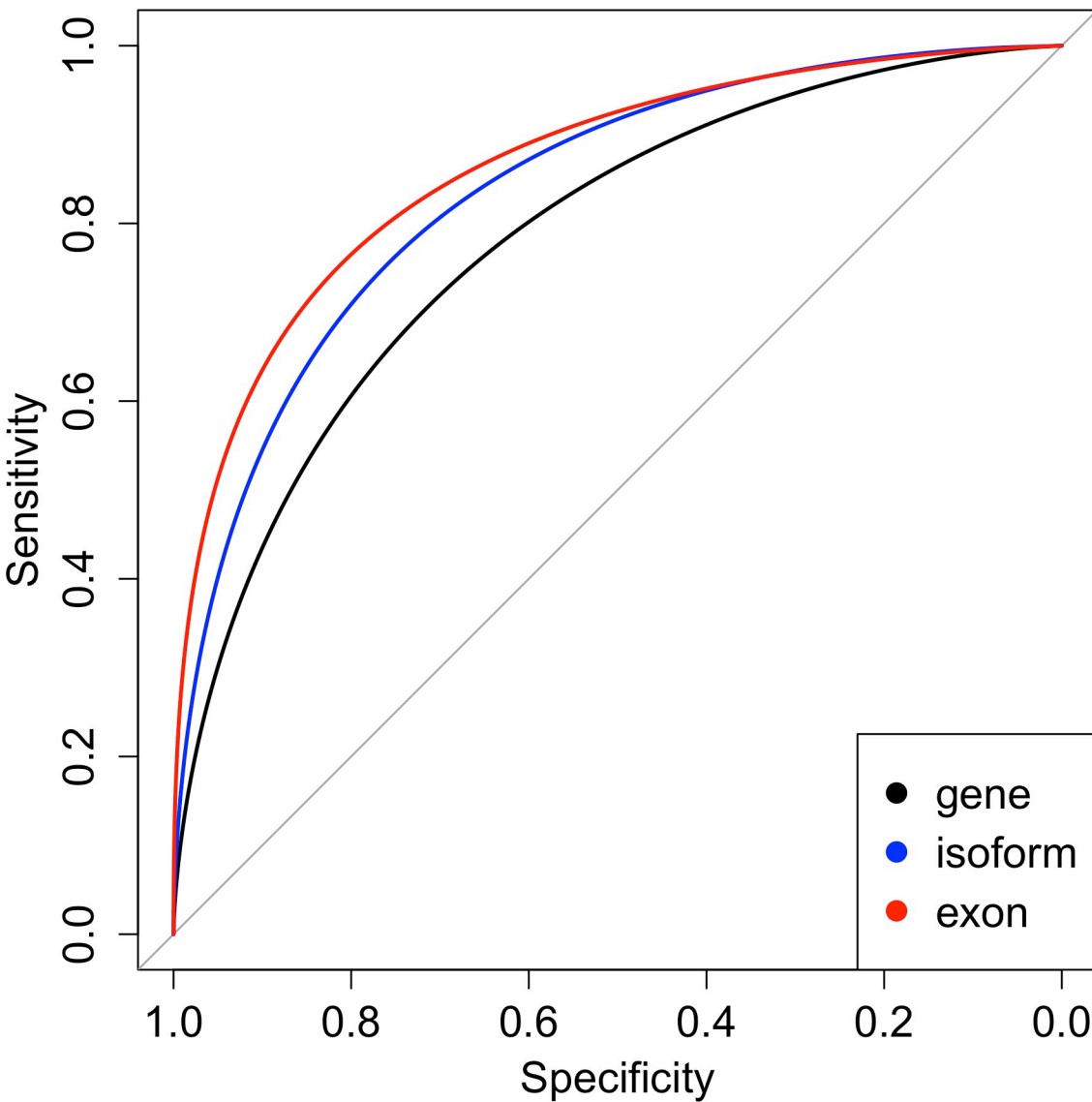

**Fig 2. ROC curves in test data for the 4-gene modified Beineke model using gene (black), isoform (blue), and exon-level (red) quantifications.** Isoform and exon-level data outperform gene-level data (Delong p = 0.002 and <0.001, respectively).

used exon level data combined with an (exon-to-)Isoform Map Layer based on curated isoform data (i.e. Ensembl GTF) and a Feature Selection Layer, as described later.

## Model optimization using a larger feature set

Having obtained improved prediction performance using exon and isoform data from four genes in the Beineke model, we then constructed models using a much larger set of features. Of the 1,270 genes that were significantly associated with current smoking from the meta-analysis by Huan et al. [2], 1,079 were expressed at levels high enough to be analyzed in our RNA-seq data. These genes contained 6,196 isoforms and 19,027 exons present in our data, and we constructed separate deep learning models using gene, isoform, and exon level data. As expected, the best models for isoform and exon data had a larger number of nodes

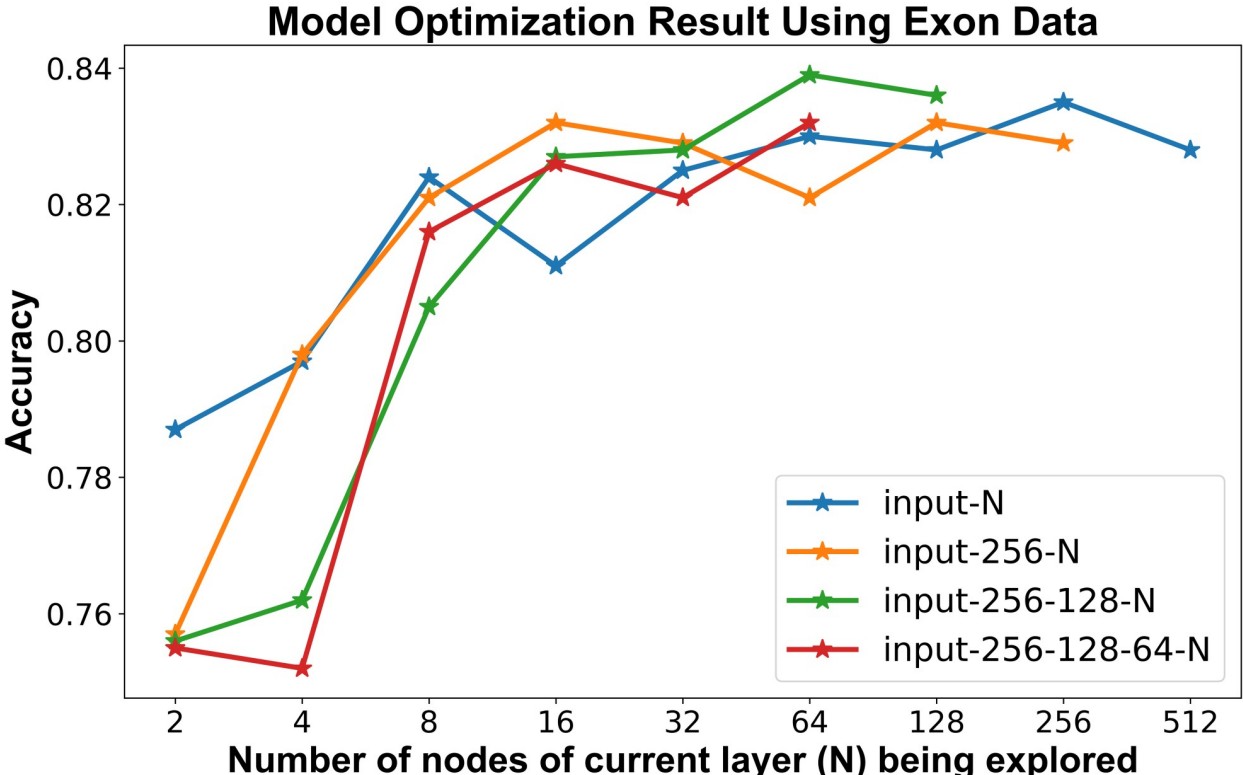

**Fig 3. Cross-validation accuracy calculated during model optimization for exon-level data.**

(256-128-64) than the gene level model (128-64-32). Maximal accuracy was observed with three layers, and the best performance was achieved with exon level quantifications (Fig 3).

### Improved prediction through isoform map and feature selection layers

We hypothesized that the performance of exon-based prediction models would be improved by incorporating relationships between exons and isoforms. Using known exon to isoform relationships from the Ensembl version 94 GTF file, we introduced a deep learning layer (IML) that encoded these connections between exons and isoforms (Fig 1), and we observed improved predictive performance in cross-validation and in test data (Table 3). For comparison, we also compared these models to models that incorporated a fully connected layer between exons and isoforms, but this model was far more complex and failed to converge. A full summary of the network architectures can be found in S4 Fig.

We then explored whether the addition of an integrated feature selection layer (FSL) would further improve performance by introducing an additional layer that assigns a non-negative weight for each input feature, and we observed an incremental increase in performance. When we compared the performance of this model to the base exon model, the performance was significantly improved ($p = 0.02$ in test data, Fig 4). However, there was no improvement from the Gene Map Layer or from providing exon and isoform quantifications directly to the Elastic Net model without any exon-to-isoform relationship information.

**Table 3. Predictive performance of various models using exon-level data, including elastic net for comparison.**

|  | Val—Accuracy | Val—AUC | Test—Accuracy | Test—AUC |
|---|---|---|---|---|
| Exon, Elastic Net | 0.821 | 0.861 | 0.774 | 0.903 |
| Exon + Iso, Elastic Net | 0.808 | 0.894 | 0.766 | 0.884 |
| Exon Base | 0.813 | 0.886 | 0.842 | 0.913 |
| Exon, GML-GTF | 0.833 | 0.899 | 0.842 | 0.913 |
| Exon, GML-GTF, FSL | 0.850 | 0.903 | 0.838 | 0.919 |
| Exon, IML-GTF | 0.843 | 0.905 | 0.854 | 0.924 |
| Exon, IML-GTF, FSL | **0.860** | **0.916** | **0.869** | **0.935** |

Val: validation data. AUC: area under the receiver operating characteristic. Exon + Iso: Concatenation of exon and isoform data. IML-GTF: Isoform Map Layer containing information from GTF file. GML-GTF: Gene Map Layer containing information from Ensembl GTF file. FSL: Feature Selection Layer. Best results are shown in bold.

## Comparison of exon-level model to serum cotinine

Cotinine is a metabolite of nicotine and the most commonly used biomarker for current smoking status. Out of 513 subjects in the test dataset, 106 had SOMAscan serum metabolite measurements available for analysis in which we could compare the performance of predictions from the exon, IML-GTF, FSL model to the discrimination performance of serum cotinine values. Interestingly, in these data the predictions from the exon-level model significantly outperformed serum cotinine (DeLong p-value = 0.01, Fig 5). The distribution of cotinine levels and exon predicted values in current and former smokers is shown in S1 and S2 Figs.

## Model interpretation

We explored the interpretation of our best performing model, (the Exon, IML-GTF, FSL model), and we generated corresponding feature importance scores for each exon using the DeepExplain [20] framework with saliency maps [21]. 48.5% of exons had non-zero scores, and the distribution of the non-zero scores was bimodal (S3 Fig). We selected the exons in the top 20% of saliency scores for gene pathway enrichment analysis using the TopGO method [22] with the 1,079 analyzed genes as the background for comparison, and 43 Gene Ontology pathways had nominally significant p-values with the most enriched pathways related to GTPase activity and protein ubiquitination/degradation. The top 10 pathways are shown in Table 4.

The most significantly enriched pathways did not have much overlap with the pathways identified in the largest previous gene-level analysis of smoking [2] or with the pathway enrichment identified in an earlier gene-level differential expression analysis in COPDGene data [3]. The most enriched pathways in these studies were primarily related to immune response, wound healing, and platelet activation. This difference is expected since the enrichment analyses performed for this study was geared to identify genes whose exon-level information was important for predicting smoking status relative to our background set of genes which consisted of the smoking-associated genes from the gene-level study of Huan et al.

## Sensitivity analyses with covariate-adjusted data

To test whether the high performance of the RNA-seq models might be tied to specific demographic characteristics of the COPDGene study population such as age, sex, or body mass index, we fitted a linear model of the expression data using these demographic covariates

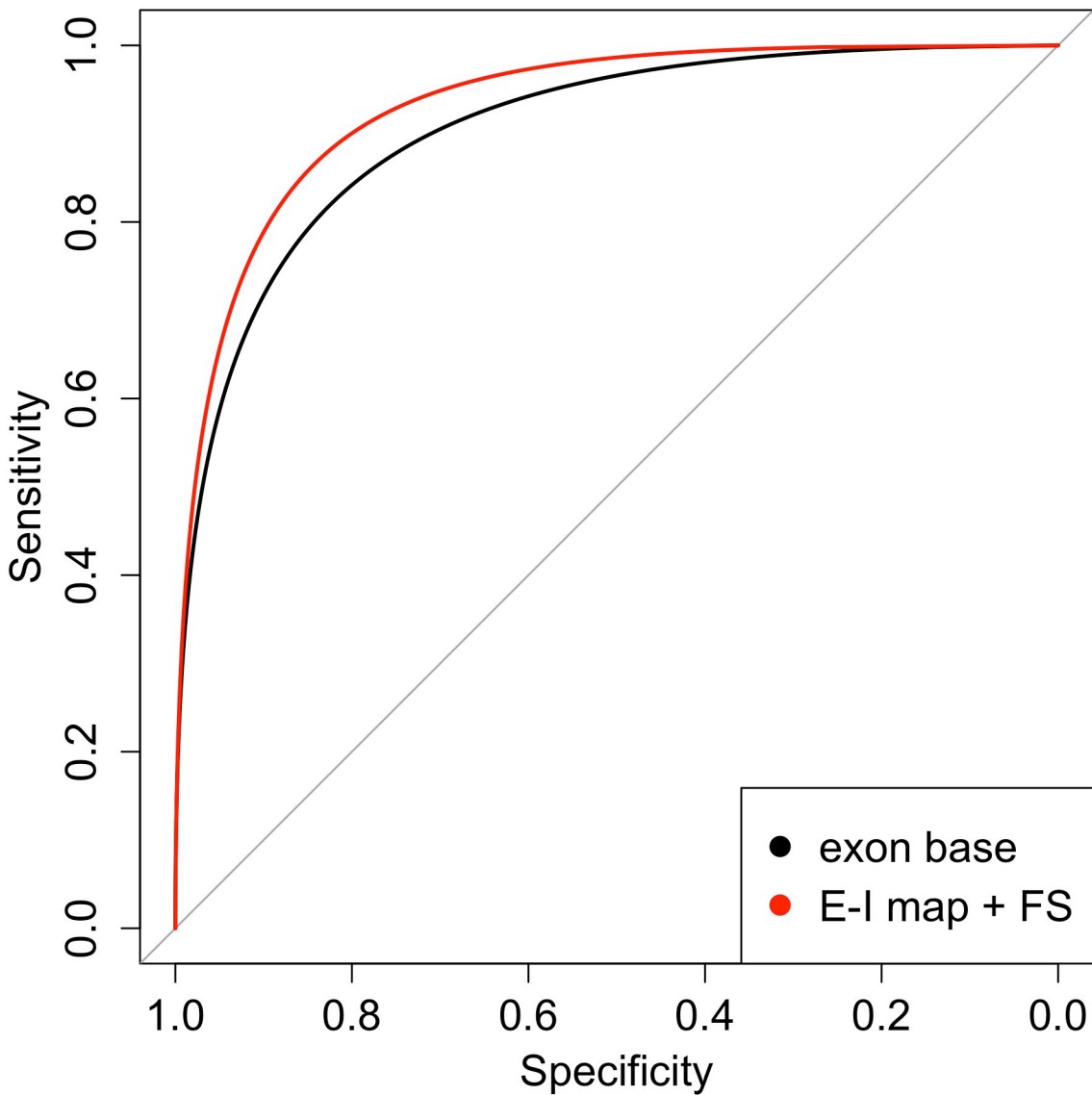

**Fig 4. ROC curves in test data for the deep learning base exon model (black) and the model including the isoform map layer and feature selection layer (red) which has significantly better performance (Delong test p = 0.02).**

together with smoking status as explanatory variables, and generated an adjusted version of the expression data by removing the effects of these covariates while retaining the main effect from smoking. We observed in the adjusted expression data a small but non-significant decrease in predictive performance for the exon-level model (AUC 0.91 versus 0.94, DeLong p-value = 0.09). These results are reported in S1E Table in S1 Text.

## Discussion

Deep learning models applied to blood RNA-seq data provide more accurate prediction of current smoking status than previously published models. In testing data, our models achieved an AUC >0.9 compared to a previously reported replication AUC of 0.81 for an established 5-gene model. Much of this improvement is due to the use of exon rather than gene expression

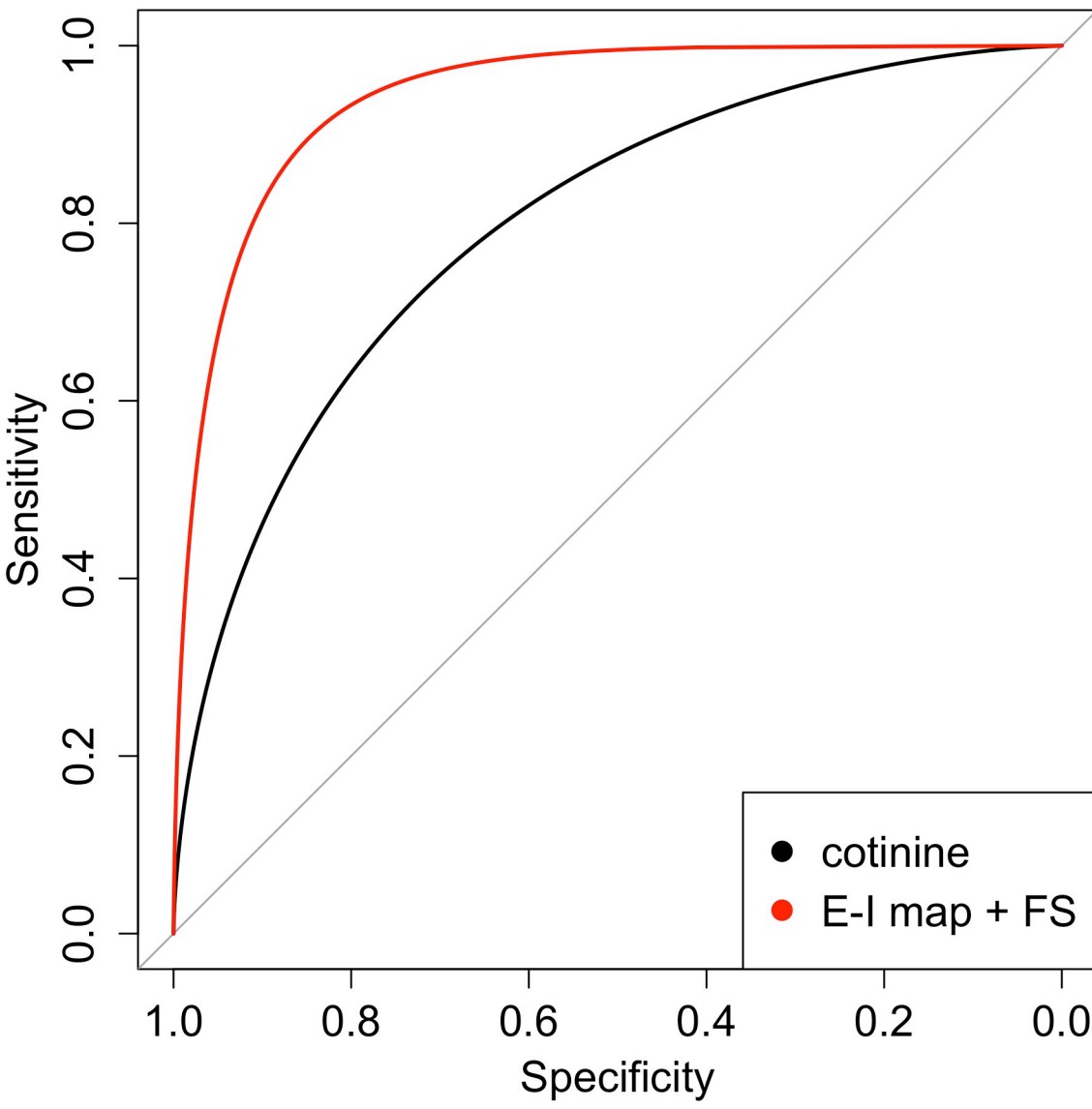

**Fig 5. ROC curves in test data for the serum cotinine (black) and the exon model including the (Exon-to-)isoform map layer and feature selection layer (red) which has significantly better performance (Delong test p = 0.01).**

levels coupled with the use of a neural net layer encoding exon to isoform relationships. These findings improve our ability to identify environmental exposures from RNA-seq data, and they suggest that latent isoform information in RNA-seq data can be used to improve clinical predictions.

This paper describes for the first time how exon and isoform-level data from RNA-seq improve the accuracy of clinical prediction models, demonstrating a general approach by which gene expression predictive models may be improved. Eukaryotic genomes are characterized by complex gene structure and extensive alternative splicing that greatly expands the protein repertoire. Over 90% of human genes have multiple transcribed isoforms [23], isoform variability is clearly observable across tissues within the same individual [24], and isoform variability is an important contributor to human diseases [25, 26]. Focusing first on the previously

**Table 4. Top 10 enriched GO pathways.**

| Go.ID | Term | Annotated | Significant | Expected | p-value |
|---|---|---|---|---|---|
| GO:0032092 | positive regulation of protein binding | 9 | 6 | 1.87 | 0.0036 |
| GO:0043547 | positive regulation of GTPase activity | 24 | 11 | 4.99 | 0.005 |
| GO:0031397 | negative regulation of protein ubiquitination | 9 | 5 | 1.87 | 0.0076 |
| GO:0006892 | post-Golgi vesicle-mediated transport | 5 | 4 | 1.04 | 0.0076 |
| GO:0006998 | nuclear envelope organization | 5 | 4 | 1.04 | 0.0076 |
| GO:0015696 | ammonium transport | 5 | 4 | 1.04 | 0.0076 |
| GO:0032722 | positive regulation of chemokine production | 5 | 4 | 1.04 | 0.0076 |
| GO:0010950 | positive regulation of endopeptidase activity | 17 | 8 | 3.53 | 0.0086 |
| GO:0032885 | regulation of polysaccharide biosynthetic process | 3 | 3 | 0.62 | 0.0089 |
| GO:0048199 | vesicle targeting, to, from or within Golgi | 3 | 3 | 0.62 | 0.0089 |

published Beineke gene expression model, we demonstrate a notable increase in performance by substituting exon or isoform quantifications for the same set of genes used in the original model (AUC increase from 0.76 to 0.86). The best performance was achieved with exon data, not estimated isoform quantifications, which is likely due to inaccuracy in the estimation of full length isoforms from short-read RNA-seq.

We were able to further improve our model by encoding known exon-isoform relationships in one of the layers of the neural network, which we refer to as the Isoform Map Layer. This is in line with other applications of machine learning to biological data that have found improved performance for algorithms that can incorporate prior biological knowledge, such as the use of known gene-interaction networks to improve the performance of clustering methods [27, 28]. Since our current catalog of human isoform variability is incomplete, as this knowledge increases the value of the Isoform Map Layer is expected to increase as well. In addition, with the growing use of long read sequencing, these more accurate isoform quantifications can be used directly as inputs to predictive models. Our data suggest that this will lead to further improvements in predictive accuracy for models based on RNA expression.

Gene expression prediction models for current smoking status are useful for multiple reasons. First, existing smoking biomarkers have good but not ideal predictive performance. In clinical practice, determination of smoking status is primarily done by patient self-report, and in instances where biochemical validation is necessary this is done via measures of nicotine metabolites, such as cotinine, in blood, urine, or saliva. While it may seem straightforward to determine smoking status, in practice it is difficult to ascertain smoking status with complete certainty for multiple reasons. Individuals may not accurately report their smoking behavior, and biochemical tests can yield false positives when individuals are exposed to nicotine in the absence of cigarette use, as can occur with the use of nicotine replacement therapy or electronic nicotine delivery devices (e-cigarettes). A systematic review of the performance of various cotinine cutoffs with respect to self-report of smoking status reported performance in the range of 70–90% sensitivity with specificity levels between 73–99% [29]. In our dataset, predictions from exon expression using an isoform mapping layer achieved a sensitivity 83% with a specificity of 89%, and when compared directly to serum cotinine levels in a subset of subjects from COPDGene, our exon expression predictive model significantly outperformed serum cotinine. However, cotinine values were only available for 106 of the test samples, and these observations may be sensitive to the method of cotinine measurement. These exploratory results require further investigation in additional datasets with standard cotinine measurements.

Another important application for transcriptome-based predictive models is to infer smoking status when only gene expression data are available. This is important because smoking has a strong effect on gene expression and therefore can be a confounder of gene expression studies, particularly in situations where smoking is confounded with specific disease states. In this scenario, use of a previously defined model to infer smoking status may allow for more accurate detection of disease-related gene expression signals, even when the smoking status of subjects has not been directly measured.

The strengths of this study are the large sample size of subjects with blood RNA-seq data, and the ability to assess our predictive models in two sets of independent test data. We assessed deep learning based methods which have provided superior predictive performance in multiple contexts, and we assessed the predictive utility of novel aspects of RNA expression which have not been extensively studied in the prediction context. Limitations of this study are that smoking status was determined by self-report only. While our models performed well in held-out test data, further validation and replication of these results in other cohorts with similar RNA isolation and sequencing protocols is necessary prior to clinical translation. Our RNA-seq libraries were generated for total RNA with globin and ribosomal RNA reduction, thus our results may not translate directly to RNA-seq data generated with poly-A selection protocols. While our study expands upon gene-level quantification studies by demonstrating the additional utility of exon-level quantification, future improvements might be obtained by considering novel, unannotated isoforms and exon junctions, rather than restricting to known transcript models as we have done here.

## Conclusion

In summary, the use of exon-level quantifications in combination with the Isoform Map Layer produced predictive models with superior ability to predict current smoking status relative to previously published models from gene expression data. These models represent the current state-of-the-art for smoking status prediction from blood RNA-seq data, and these findings are proof-of-concept that incorporating isoform level information into predictive models improves the ability to predict clinical outcomes. As the quality of isoform quantification improves from isoform inference algorithms and long-read sequencing, it is reasonable to expect that the performance of RNA-based predictive models will also improve.

## Supporting information

**S1 Text. Supplemental results.** Supplemental experiment results on upper-quartile normalized dataset and results after removing the 5 genes from the Beineke model and correlated gene partners. **S1A Table: Predictive performance of modified Beineke models using gene, isoform and exon-level expression data, with *MUC1* included and upper-quartile normalized**. Val: validation data. AUC: area under the receiver operating characteristic. IML-GTF: Isoform Map Layer containing information from Ensembl GTF file. FSL: Feature Selection Layer. Best results are shown in bold. **S1B Table: Predictive performance of modified Beineke models using gene, isoform and exon-level expression data, with *MUC1* included and TMM normalized as in the main paper**. Val: validation data. AUC: area under the receiver operating characteristic. IML-GTF: Isoform Map Layer containing information from Ensembl GTF file. FSL: Feature Selection Layer. Best results are shown in bold. **S1C Table: Predictive performance of various deep learning models using exon-level data processed with upper-quartile normalization**. Val: validation data. AUC: area under the receiver operating characteristic. IML-GTF: Isoform Map Layer containing information from Ensembl GTF file. FSL: Feature Selection Layer. Best results are shown in bold. **S1D Table: Predictive performance**

**of deep learning models using all 1079 genes, and 1020 genes with genes used in the Beineke model and their correlated genes (correlation coefficient $\geq$ 0.4) removed, and TMM normalized as in the main paper**. Val: validation data. AUC: area under the receiver operating characteristic. Base: the base architecture for gene input, Input-128-64-32-Output. Best results are shown in bold. **S1E Table: Predictive performance of modified Beineke models using gene, isoform and exon-level expression data, with *MUC1* included and covariate signals removed from the data**. Val: validation data. AUC: area under the receiver operating characteristic. IML-GTF: Isoform Map Layer containing information from Ensembl GTF file. FSL: Feature Selection Layer. Covariates removed: age, sex, body-mass index, and batch. Best results are shown in bold. **S1F Table: Predictive performance of various deep learning models using the full set of exon-level data, with covariates signals removed from the data**. Val: validation data. AUC: area under the receiver operating characteristic. IML-GTF: Isoform Map Layer containing information from Ensembl GTF file. FSL: Feature Selection Layer. Covariates removed: age, sex, body-mass index, and batch. Best results are shown in bold. (PDF)

**S2 Text. Institutional review board approval protocol numbers.** IRB protocol numbers at all participating COPDGene study sites are included as a separate Word document. (DOC)

**S1 Fig. The distribution of cotinine levels in current and former smokers.** Plasma levels of cotinine are higher in current smokers (N = 21) compared to former smokers (N = 85) in subjects from COPDGene at the second study visit. (TIF)

**S2 Fig. The distribution of exon predicted values in current and former smokers.** Predicted values from the exon model with the isoform map and feature selection layer are higher for current smokers (N = 21) than for former smokers (N = 85) in a subset of subjects from COPDGene with concurrent plasma cotinine values also available. (TIF)

**S3 Fig. Distribution of exon feature importance scores.** The distribution of log of feature importance scores for each exon using the DeepExplain [20] framework with saliency maps [21] on the trained Exon, IML-GTF, FSL model. 48.5% of exons had non-zero scores, and the distribution of the 243 non-zero scores was bimodal. The red line in the figure indicates the top 20% of exons. (TIF)

**S4 Fig. Summary of network architectures.** IML: Isoform Map Layer containing information from GTF file. GML: Gene Map Layer containing information from GTF file. FSL: Feature Selection Layer. (TIF)

## Acknowledgments

### COPDGene Investigators—Core Units

Administrative Center: James D. Crapo, MD (PI); Edwin K. Silverman, MD, PhD (PI); Barry J. Make, MD; Elizabeth A. Regan, MD, PhD

Genetic Analysis Center: Terri Beaty, PhD; Ferdouse Begum, PhD; Peter J. Castaldi, MD, MSc; Michael Cho, MD; Dawn L. DeMeo, MD, MPH; Adel R. Boueiz, MD; Marilyn G.

Foreman, MD, MS; Eitan Halper-Stromberg; Lystra P. Hayden, MD, MMSc; Craig P. Hersh, MD, MPH; Jacqueline Hetmanski, MS, MPH; Brian D. Hobbs, MD; John E. Hokanson, MPH, PhD; Nan Laird, PhD; Christoph Lange, PhD; Sharon M. Lutz, PhD; Merry-Lynn McDonald, PhD; Margaret M. Parker, PhD; Dmitry Prokopenko, Ph.D; Dandi Qiao, PhD; Elizabeth A. Regan, MD, PhD; Phuwanat Sakornsakolpat, MD; Edwin K. Silverman, MD, PhD; Emily S. Wan, MD; Sungho Won, PhD

Imaging Center: Juan Pablo Centeno; Jean-Paul Charbonnier, PhD; Harvey O. Coxson, PhD; Craig J. Galban, PhD; MeiLan K. Han, MD, MS; Eric A. Hoffman, Stephen Humphries, PhD; Francine L. Jacobson, MD, MPH; Philip F. Judy, PhD; Ella A. Kazerooni, MD; Alex Kluiber; David A. Lynch, MB; Pietro Nardelli, PhD; John D. Newell, Jr., MD; Aleena Notary; Andrea Oh, MD; Elizabeth A. Regan, MD, PhD; James C. Ross, PhD; Raul San Jose Estepar, PhD; Joyce Schroeder, MD; Jered Sieren; Berend C. Stoel, PhD; Juerg Tschirren, PhD; Edwin Van Beek, MD, PhD; Bram van Ginneken, PhD; Eva van Rikxoort, PhD; Gonzalo Vegas Sanchez-Ferrero, PhD; Lucas Veitel; George R. Washko, MD; Carla G. Wilson, MS;

PFT QA Center, Salt Lake City, UT: Robert Jensen, PhD

Data Coordinating Center and Biostatistics, National Jewish Health, Denver, CO: Douglas Everett, PhD; Jim Crooks, PhD; Katherine Pratte, PhD; Matt Strand, PhD; Carla G. Wilson, MS

Epidemiology Core, University of Colorado Anschutz Medical Campus, Aurora, CO: John E. Hokanson, MPH, PhD; Gregory Kinney, MPH, PhD; Sharon M. Lutz, PhD; Kendra A. Young, PhD

Mortality Adjudication Core: Surya P. Bhatt, MD; Jessica Bon, MD; Alejandro A. Diaz, MD, MPH; MeiLan K. Han, MD, MS; Barry Make, MD; Susan Murray, ScD; Elizabeth Regan, MD; Xavier Soler, MD; Carla G. Wilson, MS

Biomarker Core: Russell P. Bowler, MD, PhD; Katerina Kechris, PhD; Farnoush Banaei-Kashani, Ph.D

COPDGene Investigators—Clinical Centers

Ann Arbor VA: Jeffrey L. Curtis, MD; Perry G. Pernicano, MD

Baylor College of Medicine, Houston, TX: Nicola Hanania, MD, MS; Mustafa Atik, MD; Aladin Boriek, PhD; Kalpatha Guntupalli, MD; Elizabeth Guy, MD; Amit Parulekar, MD;

Brigham and Women's Hospital, Boston, MA: Dawn L. DeMeo, MD, MPH; Alejandro A. Diaz, MD, MPH; Lystra P. Hayden, MD; Brian D. Hobbs, MD; Craig Hersh, MD, MPH; Francine L. Jacobson, MD, MPH; George Washko, MD

Columbia University, New York, NY: R. Graham Barr, MD, DrPH; John Austin, MD; Belinda D'Souza, MD; Byron Thomashow, MD

Duke University Medical Center, Durham, NC: Neil MacIntyre, Jr., MD; H. Page McAdams, MD; Lacey Washington, MD

Grady Memorial Hospital, Atlanta, GA: Eric Flenaugh, MD; Silanth Terpenning, MD

HealthPartners Research Institute, Minneapolis, MN: Charlene McEvoy, MD, MPH; Joseph Tashjian, MD

Johns Hopkins University, Baltimore, MD: Robert Wise, MD; Robert Brown, MD; Nadia N. Hansel, MD, MPH; Karen Horton, MD; Allison Lambert, MD, MHS; Nirupama Putcha, MD, MHS

Lundquist Institute for Biomedical Innovationat Harbor UCLA Medical Center, Torrance, CA: Richard Casaburi, PhD, MD; Alessandra Adami, PhD; Matthew Budoff, MD; Hans Fischer, MD; Janos Porszasz, MD, PhD; Harry Rossiter, PhD; William Stringer, MD

Michael E. DeBakey VAMC, Houston, TX: Amir Sharafkhaneh, MD, PhD; Charlie Lan, DO

Minneapolis VA: Christine Wendt, MD; Brian Bell, MD; Ken M. Kunisaki, MD, MS

National Jewish Health, Denver, CO: Russell Bowler, MD, PhD; David A. Lynch, MB

Reliant Medical Group, Worcester, MA: Richard Rosiello, MD; David Pace, MD

Temple University, Philadelphia, PA: Gerard Criner, MD; David Ciccolella, MD; Francis Cordova, MD; Chandra Dass, MD; Gilbert D'Alonzo, DO; Parag Desai, MD; Michael Jacobs, PharmD; Steven Kelsen, MD, PhD; Victor Kim, MD; A. James Mamary, MD; Nathaniel Marchetti, DO; Aditi Satti, MD; Kartik Shenoy, MD; Robert M. Steiner, MD; Alex Swift, MD; Irene Swift, MD; Maria Elena Vega-Sanchez, MD

University of Alabama, Birmingham, AL: Mark Dransfield, MD; William Bailey, MD; Surya P. Bhatt, MD; Anand Iyer, MD; Hrudaya Nath, MD; J. Michael Wells, MD

University of California, San Diego, CA: Douglas Conrad, MD; Xavier Soler, MD, PhD; Andrew Yen, MD

University of Iowa, Iowa City, IA: Alejandro P. Comellas, MD; Karin F. Hoth, PhD; John Newell, Jr., MD; Brad Thompson, MD

University of Michigan, Ann Arbor, MI: MeiLan K. Han, MD MS; Ella Kazerooni, MD MS; Wassim Labaki, MD MS; Craig Galban, PhD; Dharshan Vummidi, MD

University of Minnesota, Minneapolis, MN: Joanne Billings, MD; Abbie Begnaud, MD; Tadashi Allen, MD

University of Pittsburgh, Pittsburgh, PA: Frank Sciurba, MD; Jessica Bon, MD; Divay Chandra, MD, MSc; Carl Fuhrman, MD; Joel Weissfeld, MD, MPH

University of Texas Health, San Antonio, San Antonio, TX: Antonio Anzueto, MD; Sandra Adams, MD; Diego Maselli-Caceres, MD; Mario E. Ruiz, MD; Harjinder Singh

## Author Contributions

**Conceptualization:** Zifeng Wang, Jennifer Dy, Peter J. Castaldi.

**Data curation:** Zhonghui Xu.

**Formal analysis:** Zifeng Wang, Zhonghui Xu, Peter J. Castaldi.

**Funding acquisition:** Russell Bowler, Edwin K. Silverman, Peter J. Castaldi.

**Investigation:** Zifeng Wang, Zhonghui Xu, Peter J. Castaldi.

**Methodology:** Zifeng Wang, Aria Masoomi, Jennifer Dy.

**Project administration:** Jennifer Dy, Peter J. Castaldi.

**Resources:** Russell Bowler, Edwin K. Silverman, Peter J. Castaldi.

**Software:** Zifeng Wang.

**Supervision:** Jennifer Dy, Peter J. Castaldi.

**Validation:** Zhonghui Xu, Peter J. Castaldi.

**Visualization:** Zifeng Wang, Peter J. Castaldi.

**Writing – original draft:** Zifeng Wang, Peter J. Castaldi.

**Writing – review & editing:** Aria Masoomi, Zhonghui Xu, Adel Boueiz, Sool Lee, Tingting Zhao, Russell Bowler, Michael Cho, Edwin K. Silverman, Craig Hersh, Jennifer Dy, Peter J. Castaldi.

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
