## [Decision Letter · Decision Letter 0]

17 Feb 2021

Dear Dr. Castaldi,

Thank you very much for submitting your manuscript "Improved prediction of smoking status via isoform-aware RNA-seq deep learning models" for consideration at PLOS Computational Biology.

As with all papers reviewed by the journal, your manuscript was reviewed by members of the editorial board and by several independent reviewers. In light of the reviews (below this email), we would like to invite the resubmission of a significantly-revised version that takes into account the reviewers' comments, including the call to highlight and emphasize novel biological results to a greater degree than that done in the present manuscript.

We cannot make any decision about publication until we have seen the revised manuscript and your response to the reviewers' comments. Your revised manuscript is also likely to be sent to reviewers for further evaluation.

Best regards,

Donna K. Slonim

Associate Editor

PLOS Computational Biology

Florian Markowetz

Deputy Editor

PLOS Computational Biology

Reviewer's Responses to Questions

**Comments to the Authors:**

Reviewer #1: Please see the attached file for my comments.

Reviewer #2: The authors present a deep learning model that predicts smoking status based on blood RNA-seq data. The model uses exon and isoform-level features, which are shown to improve accuracy over gene-only features. The models are trained on a new and extremely large set of blood RNA-seq data from 2,557 subjects, which is made publicly available in GEO.

The new data made available with this manuscript is arguably its biggest contribution and I commend the authors for making these data publicly available. I am less convinced by the contribution of a deep learning model for smoking status for a variety of reasons outlined in my specific comments below.

1. The model, particularly the deep net with the larger set of features, could actually be picking up on features that are directly associated with covariates (e.g., age, sex, BMI), which could then have associations with smoking status. In other words, the model may be predicting these covariates, not smoking status directly, which is dangerous when applying the model to other populations. The authors need to show that the model is truly learning expression signatures that are directly related to smoking, and not these covariates.

2. Related to 1, the authors do not characterized the features that the learned models are picking up on and what biology these might suggest with respect to smoking status. For a paper to PLoS CompBio, I would expect more characterization of the molecular biology.

3. I'm not fully convinced that the deep net is outperforming a simple logistic regression model with similar features. In Table 3, I believe the "Elastic Net" model is a logistic regression model. If this model is also given isoform abundances (i.e., Exon + Isoform, Elastic Net), what is the performance? The IML layer is essentially giving isoform level information to the deep net, so for a fair comparison, this information should also be given to the logistic regression model. One really needs to show a big gain in performance with a deep net over a simpler model to justify its use, and I am not seeing such a difference here.

4. In the authors' previous study [3], they identified differentially expressed exons (while taking into account covariates!) associated with smoking status. Why are those not used in this study?

**Have all data underlying the figures and results presented in the manuscript been provided?**

Reviewer #1: **No: **See my comments regarding the exon/gene definitions, model weights, and source code. The authors should also upload their test set predictions.

Reviewer #2: Yes

PLOS authors have the option to publish the peer review history of their article (what does this mean?). If published, this will include your full peer review and any attached files.

Reviewer #1: No

Reviewer #2: No
---

## [Decision Letter · Decision Letter 1]

21 Jun 2021

Dear Dr. Castaldi,

Thank you very much for submitting your manuscript "Improved prediction of smoking status via isoform-aware RNA-seq deep learning models" for consideration at PLOS Computational Biology.

As with all papers reviewed by the journal, your manuscript was reviewed by members of the editorial board and by the independent reviewers.  The reviewers are happy with most of the changes but have identified a few essential points that still raise questions about the generality of the results and improving readers' understanding of the biological insights provided.   In light of the reviews (below this email), we would like to invite the resubmission of a further revised version that takes into account the reviewers' comments.  

We note that the journal's data and code-sharing policy allows GitHub as a repository, although it does encourage further archiving the GitHub submission via Zenodo.   Questions about the completeness of what is in GitHub for reproducibility purposes, however, are definitely relevant.   Please examine the journal policy pages carefully regarding availability.  

We cannot make any decision about publication until we have seen the revised manuscript and your response to the reviewers' comments. Your revised manuscript is also likely to be sent to reviewers for further evaluation.

Sincerely,

Donna K. Slonim

Associate Editor

PLOS Computational Biology

Florian Markowetz

Deputy Editor

PLOS Computational Biology

Reviewer's Responses to Questions

**Comments to the Authors:**

Reviewer #1: Please see the attached file for my comments.

Reviewer #2: The authors have sufficiently addressed my prior comments. I have two remaining minor comments on the revised manuscript.

1. The additional work on model interpretation strengthens this manuscript. However, the biological findings (e.g., top pathways related to GTPase activity and protein ubiquitination/degradation) are not connected to any of the findings from prior related work. It would be helpful if the authors could comment on whether these findings reinforce prior findings or are novel by citing prior work.

2. It is excellent that the authors have made all of the code and supporting files available. However, some of the key files (e.g., the isoform-exon maps and network weights) are only available via a Google drive link. Files in Google drive can be easily inadvertently moved or deleted. I would strongly suggest archiving and versioning these key files on a site such as Zenodo to ensure reproducibility of this work well into the future.

**Have the authors made all data and (if applicable) computational code underlying the findings in their manuscript fully available?**

Reviewer #1: **No: **Please see my comments regarding the exon/gene definitions, model weights, and source code. These data should be made available on an open-access archive such as Dryad or Zenodo.

Reviewer #2: Yes

PLOS authors have the option to publish the peer review history of their article (what does this mean?). If published, this will include your full peer review and any attached files.

Reviewer #1: No

Reviewer #2: No
---

## [Decision Letter · Decision Letter 2]

8 Sep 2021

Dear Dr. Castaldi,

We are pleased to inform you that your manuscript 'Improved prediction of smoking status via isoform-aware RNA-seq deep learning models' has been provisionally accepted for publication in PLOS Computational Biology.

Please note that your manuscript will not be scheduled for publication until you have made the required changes, so a swift response is appreciated.  We would suggest that at this stage, you also look at the one remaining comment (#7) from Reviewer 1 about L1 regularization and determine whether the suggested very minor text edit would further clarify the presentation.    

Best regards,

Donna K. Slonim

Associate Editor

PLOS Computational Biology

Florian Markowetz

Deputy Editor

PLOS Computational Biology

Reviewer's Responses to Questions

**Comments to the Authors:**

Reviewer #1: Reviewer comments uploaded as an attachment.

Reviewer #2: The authors have sufficiently addressed my previous comments.

**Have the authors made all data and (if applicable) computational code underlying the findings in their manuscript fully available?**

Reviewer #1: Yes

Reviewer #2: Yes

PLOS authors have the option to publish the peer review history of their article (what does this mean?). If published, this will include your full peer review and any attached files.

Reviewer #1: No

Reviewer #2: No

---

## [Editor Report · Acceptance letter]

27 Sep 2021

PCOMPBIOL-D-20-02234R2

Improved prediction of smoking status via isoform-aware RNA-seq deep learning models

Dear Dr Castaldi,

I am pleased to inform you that your manuscript has been formally accepted for publication in PLOS Computational Biology. Your manuscript is now with our production department and you will be notified of the publication date in due course.

With kind regards,

Andrea Szabo
